Optimizing soil health through activated acacia biochar under varying irrigation regimes and cultivars for sustainable wheat cultivation

Komal Lubaba 1
Jahan Summera summera.botany@pu.edu.pk 1
Kamran Atif 1
Hashem Abeer 2
Avila-Quezada Graciela Dolores 3
Abd_Allah Elsayed Fathi 4
1 Institute of Botany, University of the Punjab , Lahore , Pakistan
2 Botany and Microbiology Department, College of Science, King Saud University , Riyadh , Saudi Arabia
3 Facultad de Ciencias Agrotechnologicas, Universidad Autonoma de chihuahua , Chihuaha , Mexico
4 Plant Production Department, College of Food and Agricultural Sciences, King Saud University , Riyadh , Saudi Arabia
Abd El-Moneim Diaa
Electronic publication date: 2025 Jan 17
Publication date: 2025
Volume: 13
Electronic Location ID: e18748
Received 2024 Oct 15; Accepted 2025 Dec 2
Copyright: ©2025 Komal et al.
Copyright year: 2025
Copyright holder: Komal et al.
License: This is an open access article distributed under the terms of the Creative Commons Attribution License, which permits unrestricted use, distribution, reproduction and adaptation in any medium and for any purpose provided that it is properly attributed. For attribution, the original author(s), title, publication source (PeerJ) and either DOI or URL of the article must be cited.
License URL: https://creativecommons.org/licenses/by/4.0/

Keywords: Activated biochar, Water scarcity, Antioxidants, Soil microporosity, Wheat yield, Photosynthetic pigments, Organic matter

Funding: Higher Education Commission (HEC) of Pakistan providing funds for NRPU research project 20-16716 Researchers Supporting Project Number, King Saud University, Riyadh, Saudi Arabia RSP2025R134 This study was funded by Higher Education Commission (HEC) of Pakistan providing funds for NRPU research project (Grant Number 20-16716) and Researchers Supporting Project Number (RSP2025R134), King Saud University, Riyadh, Saudi Arabia. The funders had no role in study design, data collection and analysis. The funder played a direct role in the decision to publish, and preparation of the manuscript.

==============================
Wheat, a staple food crop globally, faces the challenges of limited water resources and sustainable soil management practices. The pivotal elements of the current study include the integration of activated acacia biochar (AAB) in wheat cultivation under varying irrigation regimes (IR). A field trial was conducted in the Botanical Garden, University of the Punjab, Lahore during 2023–2024, designed as a split-split-plot arrangement with RCBD comprising three AAB levels (0T, 5T, and 10T, T = tons per hectare) three wheat cultivars (Dilkash-2020, Akbar-2019, and FSD-08) receiving five IR levels (100%, 80%, 70%, 60%, and 50% field capacity). Biochar amended soil showed improved BET surface area, pore size, and volume. Carbon recovery (45%) and carbon sequestration capacity (49%) of 10T-AAB amended soil were better than non-amended soil (0.43% and 0.13%, respectively). The 10T-AAB amendment significantly improved the soil’s microporosity and water retention capacity, increasing it by 1.1 and 2.2 times, respectively. Statistical analysis showed that a reduction in IR negatively affected plant growth and yield. The 10T-AAB levels significantly increased sugar contents (14%), relative water content (10–28%), membrane stability index (27–55%), and photosynthetic pigments (18–26%) of wheat leaves under deficit irrigation among all the cultivars. Maximum stress markers (catalase, proline, peroxidase, and superoxide dismutase) were observed from Akbar under 50% irrigation with 0T-AAB, and the least were observed from 50% irrigated Dilkash-2020 with 10T-AAB amended soil. Among cultivars, Dilkash-2020 was observed to be the best for maximum yield, followed by FSD-08 and Akbar-2019, respectively. When compared to other IR levels, 10T-AAB amended soil had the highest yield enhancement (12, 11, and 9.2 times for Dilkash-2020, FSD-08, and Akbar-2019, respectively). Hence, AAB enhanced wheat production by improving soil properties, drought resilience, and yield attributes.

Introduction

Water scarcity, driven by climate change, is altering evapotranspiration patterns, soil moisture level, and plant rhizosphere dynamics (Albacete, Martínez-Andújar & Pérez-Alfocea, 2014). Addressing these challenges requires targeted specific crop management strategies that support water availability during stress intervals while enhancing crop productivity (El Chami et al., 2019). Such strategies can boost crop water use efficiency, reduce surface runoff, and limit deep percolation losses (Capraro et al., 2018). In regions where fertile soils and water resources are limited, improving wheat production (a globally crucial crop) is essential for food security (Huang et al., 2022). Wheat faces ongoing challenges from both water scarcity and the need for sustainable soil management practices (Yu et al., 2020).

Biochar has gained attention as an amendment that enhances soil’s physicochemical properties and moisture retention (Shakeel et al., 2022; Jahan et al., 2022). Biochar is produced through pyrolysis of organic substances at a very high temperature; it offers soil a high surface area and micropore volume, crucial for moisture retention and water conservation (Lehmann & Joseph, 2015). Activation of biochar, particularly organic methods like vermicompost or perlite addition, is reported to enhance these benefits further (Sanchez-Hernandez, Ro & Díaz, 2019). The combination of biochar with perlite and vermicompost supports improved water retention through a combined effect of hydrated volcanic glass and decomposing organic matter, fostering a more resilient rhizosphere (Jahan et al., 2022).

Activated biochar contributes to soil structure, water retention, and favorable plant morpho-physiological and biochemical responses (Lehmann & Joseph, 2015). It aids in soil health recovery, enhanced water use efficiency, and sustained plant growth, even under water-limited conditions (Iqbal et al., 2024). Additionally, biochar’s log-term carbon retention enhances carbon sequestration, promoting sustainable soil health and productivity (Daer et al., 2024). Such improvements in soil conditions support plant water uptake, reduce reactive oxygen species (ROS) production, and stabilize photosynthetic pigments and sugar contents under stress (Wu et al., 2023). Consequently, biochar helps mitigate water-deficit stress impacts by increasing antioxidant enzyme activity, such as peroxidases and superoxide dismutase (Shakeel et al., 2022). Various field studies are increasingly documenting the positive effects of biochar; however, plant responses significantly vary depending on the type of biochar, activation methods, and biomass properties (Jahan et al., 2024).

Despite its proven benefits, field studies on biochar application, especially activated biochar, are limited in the context of deficit irrigation for wheat production. Exploring its integration with precision irrigation could lead to enhanced crop resilience and productivity. This study fills the knowledge gap by investigating activated biochar’s role in supporting wheat under deficit irrigation, contributing to soil health, and sustaining agricultural productivity. Given these conditions, we hypothesize that soil amended with activated biochar will support wheat growth and productivity under deficit irrigation conditions by improving soil moisture retention and enhancing soil physicochemical properties. The primary objective of this study is to assess the synergistic effects of activated biochar on soil quality, plant physiology, growth, and yield in three commercial wheat cultivars (Dilkash-2020, Akbar-2019, and FSD-08) under variable irrigation regimes. This study will help to identify the optimal biochar amendment and irrigation regime combination for water-scarce areas, promoting sustainable agriculture and food security. This research is novel in that it examines organically activated biochar tailored to wheat production under water-stressed conditions and evaluates their potential in practical field settings. The findings aim to bridge critical gaps in sustainable crop management by developing our understanding of activated biochar’s role in improving water stress resilience in wheat.

Materials and Methods

Production of activated biochar

Wood twigs of Acacia nilotica were utilized for biochar production as optimized by Jahan et al. (2022). Before pyrolysis, raw biomass was air-dried to reduce its moisture content. Production of biochar was carried out by the slow pyrolysis technique at 450 °C for a three-hour duration using a batch pyrolysis temperature-controlled unit. After the cooling, the physico-chemical properties of biochar were analyzed by Jahan et al. (2023). For activation purposes, biochar, vermicompost, and perlite were mixed in a 1:1:1 ratio along with molasses to speed up the process and incubated for thirty days. Mixing and turning of the material was done daily to maintain proper aeration. After incubation, samples of the activated acacia biochar were assessed to determine its physicochemical characteristics (Jahan et al., 2023).

Experimental design and area

A field trial was executed at Botanical Garden, University of the Punjab, Lahore, Pakistan (N31°30′4.3236″, E74°18′5.4684), during 2023–2024. The experiment comprised of split-split plot arrangement with a randomized complete block design (RCBD) in triplicate with a plot size of 5.95 m2. Factors under observation comprised of activated acacia biochar (0T-AAB, 5T-AAB, and 10T-AAB), cultivars (Dilkash-2020, Akbar-2019, and FSD-08), and irrigation regimes (100%, 80%, 70%, 60%, and 50% field capacity). Activated acacia biochar (AAB) was applied manually to the topsoil (15 cm) and thoroughly mixed as 2.7 and 5.4 kg per plot for 5 and 10 tons per hectare, respectively. Cultivars were selected as per recommended cultivars for irrigated soils from the Ayub Agricultural Research Institute (AARI), Faisalabad. Basal fertilizer doses for nitrogen, phosphorous, and potassium were applied in the form of urea (20.4 g per plot), SOP (36 g per plot), and DAP (54 g per plot), but urea was applied in two splits with a second dose in subsequent irrigation.

Meteorological data

Meteorological data was obtained from EOS data analytics: Crop Monitoring System (https://crop-monitoring.eos.com/). Parameters of meteorological data specifically included, minimum and maximum temperature (°C), wind speed (m/s), specific humidity (%), and precipitation (mm) (Fig. 1).

Figure 1 Crop meteorological data retrieved from crop monitoring system database presenting variation in Max Deg.C (maximum temperature in °C), Min Deg.C (minimum temperature in °C), wind speed (m/s), specific humidity (m/s), humidity (%), and precipitation (mm).

Strategy for maintaining irrigation regimes

Water requirements of crop was calculated by the Eq. (1) presented by the Food and Agriculture Organization (FAO). (1) IN=ETc−Pe

where IN presents the net water requirement, ETc stands for evapotranspiration of crop, and Pe shows effective rainfall. Moreover, evapotranspiration was estimated by using the expression (Eq. 2) as given by Mehta & Pandey (2015), (2) ETc=ETo×Kc

where Eto is reference evapotranspiration and Kc is crop coefficient. Reference evapotranspiration was determined by using Penman Monteith Eq. (3) Mehta & Pandey (2015), (3) ETo=0.14ΔRn−G+γ900T+273U2es−eaΔ+γ1+0.34U2

where T is the daily mean temperature (°C) at a height of 2 m, Rn indicates net radiation, G presents the soil heat flux in MJm2/day, Δ presents the gradient of the vapor pressure-temperature curve in KPa/°C, γ is the psychometric constant (KPa/°C), U2 shows the wind rate per day at a 2 meter elevation in meters per second, and es and ea present the average and real saturation vapor pressure, respectively. Reference evapotranspiration (ETo) was calculated by CROPWAT 8.0 (Soomro et al., 2023). The effective rainfall was observed as given below (Eq. 4). (4) Pe=0.6×P−3.33.

If P ≤ 70 mm, Pe and P show effective rainfall and total precipitation respectively. Irrigation regimes (80% to 50%) were calculated using the percentage formula, based on evapotranspiration patterns. Soil moisture content was observed using a moisture meter (Lutron PMS-714) at regular intervals before each irrigation to estimate the field capacity.

Soil and biochar physicochemical analysis

Soil pH and electrical conductivity were estimated using a pH and EC meter by following the standard procedure of Rayment & Lyons (2011). Standard procedures by Estefan, Sommer & Ryan (2013) were followed for estimation of water holding capacity, soil porosity, and pore size. The yield of activated biochar was assessed using Eq. (5). Brunauer-Emmett-Teller (BET) and Barrett-Joyner-Halenda (BJH) were used for soil particle surface area analysis, pore size and volume analysis using Quantachrome Instruments version 11.04 with nitrogen gas media. Carbon recovery (CR) was estimated using Eq. (6) (Li et al., 2022). Mean residence time (MRT) and carbon (%) remaining in soil over 100 years (HC+100), were calculated according to Eqs. (7) and (8), respectively (Venkatesh et al., 2022). H/C shows the atomic ratio of activated biochar and amended soils. The letter ‘e’ represents an exponential term. As shown in Eq. (9), R50 presents an indicator of carbon’s recalcitrance in amended soil and activated biochar (Harvey et al., 2012; Li et al., 2022), whereas, T50Graphite and T50Biochar are temperatures required for 50% weight loss of activated graphite and biochar, respectively. Graphite was used as reference substance with purity ≥ 99.85% and 100 mesh. Equation (10) was used to estimate the carbon sequestration potential of activated biochar (Venkatesh et al., 2022). (5) Yield=BiocharweightRawweight×100

(6) CarbonRecoveryCR=CBiocharCBiomass×Yield

(7) MRT=4501×e−3.2×HC

(8) HC+100=1.05−0.616×HC

(9) R50=T50BiocharT50Graphite

(10) Carbonsequestration=R50×CR.

Plant physiological and biochemical analysis

Wheat leaves were analyzed for leaf proline content at the grain filling stage by following Bates, Waldren & Teare (1973). The method of DuBois et al. (1956) was used to evaluate the sugar contents in the leaf sample. Lipid peroxidation in the leaf sample was analyzed by the method of Procházková, Boušová & Wilhelmová (2011) where malondialdehyde (MDA) was the indicator of lipid peroxidation. The membrane stability index (MSI) of the leaf was assessed by method given by Sairam (1994). The method of Mullan & Pietragalla (2012) was followed for relative water content (RWC). The Arnon method was used to find the chlorophyll content (Arnon, 1949) while carotenoid content was assessed by the method of Lichtenthaler & Wellburn (1983). Protein content was estimated by the method of Bradford (1976). Beauchamp & Fridovich (1971) method was used to observe the activity of superoxide dismutase (SOD). The peroxidase (POD) level was analyzed following the method of Gorin & Heidema (1976) and the method of Iwase et al. (2013) was used to analyze the catalase activity in the leaf sample.

Plant growth and yield analysis

Plant growth and yield parameters were analyzed at the grain filling stage and a digital analytical balance (Model FA2204E, China) was used to measure the fresh and dry weights. The method of Usman, Liedl & Shahid (2014) was used to determine apparent water productivity as follows: (11) ApparentwaterproductivityKgm−3=SeedYieldkgha−1IrrigationWaterm−3.

Statistical analysis

Experimental data was analyzed using IBM SPSS Statistics 23.0 software for analysis of variance (ANOVA) and post-hoc comparisons, including Duncan’s test for alphabetic arrangement of data ranges. Descriptive statistics were used for generating graphs based on means and standard deviation. Pearson correlation was generated through Origin 2024 software (https://www.originlab.com/2024). The PCA analysis and heatmap were constructed to predict the correlation of treatments with growth variables of wheat grown under varying irrigation regimes using RStudio (R-4.3.1-x86 64.pkg) (R Core Team, 2023; RStudio Team, 2023).

Results

Soil and biochar physicochemical analysis

Table 1 presents the physicochemical soil analysis. The organic matter (OM) was highest in soil with 10T-AAB amendment, reaching 5.2%. There was 1.25 fold higher organic carbon in soils amended with 10T-AAB as compared to 0T-AAB amendment. The highest percent carbon value was observed by 10T amended soil with 3% carbon, whereas nitrogen content peaked in 5T-AAB amendment with 1.07%. Soil water holding capacity (WHC) was at its maximum in 10T-AAB amended soil with 28%, which is 37% higher than non-amended soil. Macropore space was highest at 118% under 0T-AAB amendment. Porosity was highest in 10T-AAB amended soil, at 269.38 (1.1 folds higher). The pH levels remained stable, peaking at 6.7, i.e., near to neutral pH and effective for wheat growth in 10T-AAB amended soil. The 10T-AAB amended soil showed slightly increased electrical conductivity (EC). Other attributes, including hydrogen, oxygen, carbon recovery, carbon sequestration capacity, and mean residence time, were increased in 10T-AAB amended soil (Table 1).

Table 1 Physicochemical properties of activated acacia biochar (AAB), non-amended soil (0TAAB AS), 5 tons per hectare (5T-AAB AS) and 10 tons per hectare (10T-AAB AS) amended soil.

Traits	AAB	0T-AAB AS	5T-AAB AS	10T-AAB AS	
C	63.78a	2.31d	2.61c	2.90b	
H	3.20a	0.25c	0.28b	0.31b	
O	12.04d	19.47a	19.34b	17.10c	
N	0.76d	0.83c	1.07a	1.04b	
H/C	0.60b	1.35a	0.42c	0.35d	
O/C	0.14d	6.33a	5.56b	4.42c	
Ash	13.50a	9.65d	10.08c	10.26b	
Volatile Matter	33.70a	3.26d	5.83c	7.27b	
Organic matter (%)	53.30a	4.15d	4.69c	5.22b	
WHC (%)	92.73a	20.43d	24.03c	27.93b	
Pore Space (%)	98.29d	117.58a	115.58b	110.58c	
Porosity (%)	429.75a	244.98d	254.29c	269.37b	
BET SA (m2 g-1)	8.390c	4.360d	20.730a	19.136b	
BJH PS (Å)	15.158c	19.732a	15.753b	15.078d	
BJH PV (cc g-1)	0.003c	0.003c	0.007a	0.006b	
Ph	6.92a	5.30d	6.10c	6.70b	
EC	1.27a	0.64d	0.76c	0.84b	
Yield (%)	53.36a	8.95d	39.24c	46.48b	
CR (%)	70.56a	0.43d	34.10c	44.97b	
CS (%)	25.61c	0.13d	36.46b	48.78a	
MRT (y)	663.45c	59.45d	1172.18b	1478.29a	
HC+100 (%)	68.04c	21.84d	79.13b	83.44a	
Notes.

C = Carbon, H = Hydrogen, O = Oxygen, N = Nitrogen, H/C= Hydrogen to carbon ratio, O/C= Oxygen to carbon ratio, EC= Electrical conductivity, SA= surface area, PS = Pore size, PV= Pore volume, CR = Carbon recovery, CS = Carbon Sequestration, MRT = Mean residence time, HC+100 = the percent of the carbon that would remain in the soil after 100 years and the unit of MRT in the table is year (y), Data presenting Mean and various superscripted alphabets indicate statistical significance at 95% confidence interval. AAB= activated acacia biochar, 0T-AAB AS= non amended soil, 5T-AAB AS = 5 tons per hectare activated acacia biochar amended soil, and 10T-AAB AS = 10 tons per hectare activated acacia biochar amended soil.

Figure 2 Mean comparison for the effect of varying irrigation regimes, AAB and cultivars.

(A) Proline (mg g−1), (B) sugar contents (mg g−1), (C) malondialdehyde (pmol ml−1), and (D) relative water content. Lowercase letters indicate statistical significance at 95% confidence interval, 0T, 0 tons per hectare; 5T, 5 tons per hectare; 10T, 10 tons per hectare; AAB, activated Acacia Biochar; and IR, irrigation regime; Dilkash, Dilkash-2020 cultivar; Akbar, Akbar-2019 cultivar; and FSD-08, FSD-08 cultivar.

Plant physiological and biochemical attributes

The analysis of variance showed a significant effect of activated acacia biochar (AAB) on the proline content of wheat cultivars under varying irrigation regimes. The mean comparison showed that 10T-AAB reduced proline content by 48%, 58%, and 39% in Dilkash-2020, Akbar-2019, and FSD-08, respectively, in 50% IR when compared to 0T-AAB (Fig. 2A). Sugar contents were reduced with a reduction in IR by 7%, 8%, and 14% in Dilkash-2020, Akbar-2019, and FSD-08, respectively, in 50% IR as compared to 100% irrigated plants in 0T-AAB (Fig. 2B). The AAB significantly reduced the MDA content under deficit IR (Fig. 2C). Results showed that AAB amendment significantly improved RWC, MSI, and other physiological attributes of wheat under varying irrigation regimes. Compared to control (0T-AAB), 5T-AAB and 10T-AAB increased RWC by 10% and 28%, respectively, in Dilkash-2020 (Fig. 2D).

Deficit IR reduced MSI by 20–50% with 0T-AAB. Whereas 5T-AAB and 10T-AAB improved overall MSI by 27% and 55%, respectively (Fig. 3A). The significant effect of biochar on photosynthetic pigments were observed under deficit IR. Compared to control (0T-AAB), 5T-AAB and 10T-AAB increased Chl a content by 18 and 26 times, respectively (Fig. 3B). whereas 5T-AAB and 10T-AAB produced Chl b content 18 and 27 times higher, respectively (Fig. 3C) and maximum carotenoids content was observed from 10T-AAB with 100% IR in Dilkash-2020 (Fig. 3D).

Figure 3 Mean comparison for the effect of varying irrigation regimes, AAB and cultivars.

(A) Membrane stability Index (%), (B) chlorophyll a (mg g−1 FW), (C) chlorophyll b (mg g−1 FW), and (D) carotenoids (mg g−1 FW) in fresh leaves of wheat at grain filling stage. Lowercase letters indicate statistical significance at 95% confidence interval, 0T, 0 tons per hectare; 5T, 5 tons per hectare; 10T, 10 tons per hectare; AAB, activated Acacia Biochar; and IR, Irrigation regime; Dilkash, Dilkash-2020 cultivar; Akbar, Akbar-2019 cultivar; and FSD-08, FSD-08 cultivar.

Biochar amendment in low IR was observed to increase the protein contents in all cultivars but the major increase (18 times higher) was observed in Dilkash-2020 with 10T-AAB in 70% IR as compared to its counterpart with 0T-AAB followed by 50% IR with 10T-AAB (Fig. 4A). For catalase (Fig. 4B) and peroxidase (Fig. 4C), the peak level was in Akbar-2019 at 50% IR with 0T-AAB. Superoxide dismutase had the highest value in Akbar-2019 at 50% IR as well (Fig. 4D). The AAB amendment in deficit irrigation reduced the antioxidant activity by decreasing these enzymes levels by 17–57% in all cultivars.

Figure 4 Mean comparison for the effect of varying irrigation regimes, AAB and cultivars.

(A) Protein (mg g−1) (B) catalases (units/mg protein), (C) peroxidases (units/mg protein), and (D) superoxide dismutase (units/mg protein) in fresh leaves of wheat at grain filling stage; lowercase letters indicate statistical significance at 95% confidence interval 0T, 0 tons per hectare; 5T, 5 tons per hectare; 10T, 10 tons per hectare; AAB, activated Acacia Biochar; and IR, Irrigation regime; Dilkash, Dilkash-2020 cultivar; Akbar, Akbar-2019 cultivar; and FSD-08, FSD-08 cultivar.

Plant growth and yield attributes

There was a significant (p ≤ 0.05) effect of AAB on plant growth indices and irrigation regimes significantly affected plant growth except for the number of tillers and leaves. The difference between cultivars for the number of tillers and root length was not significant. Plant morphological traits including leaf fresh weight, stem fresh weight, root fresh weight, leaf dry weight, stem dry weight, and root dry weight showed significant increases with biochar (5T-AAB and 10T-AAB), and percent increases ranged from 16% to 81%, respectively, compared to the control (0T-AAB) under deficit IR conditions (Tables 2 & 3). It was observed that reduction in IR significantly decreased plant yield by 77%, 81%, and 82% in Dilkash-2020, FSD-08, and Akbar-2019, respectively (Table 4). But when these cultivars were grown in amendment with AAB, increased yield attributes were observed with both 5T-AAB (114%, 112%, and 88%, respectively) and 10T-AAB (119%, 110%, and 92%, respectively). The highest yield was observed from 100% IR with 10T-AAB in Dilkash-2020 and Akbar-2019 cultivars, but 5T-AAB in 70% IR gave the best grain yield for FSD-08. For spike length, spike weight, number of spikes per plant, spikelet per spike, and grains per spike, 10T-AAB in 70% IR proved to be the best (Table 4). Maximum yield per hectare and highest apparent water productivity were observed from Dilkash-2020 with 10T-AAB in 100%, followed by 70% IR (Figs. 5A and 5B).

Table 2 Means comparison for the effect of varying irrigation regimes, AAB levels of amendment and cultivars on leaf fresh weight (g), stem fresh weight (g), root fresh weight (g), leaf dry weight (g), stem dry weight (g) of wheat.

		Leaf fresh weight (g)	Stem fresh weight (g)	Root fresh weight (g)	Leaf dry weight (g)	Stem dry weight (g)	
Cult.	IR	0T-AAB	5T-AAB	10T-AAB	0T-AAB	5T-AAB	10T-AAB	0T-AAB	5T-AAB	10T-AAB	0T-AAB	5T-AAB	10T-AAB	0T-AAB	5T-AAB	10T-AAB	
Dilkash-2020	100%IR	5.13 ± 0.15c	6.95 ± 1.27ef	5.39 ± 0.2b	16.30 ± 0.70ef	22.93 ± 0.09b	22.96 ± 1.03b	4.84 ± 0.60c	4.92 ± 0.40bc	6.19 ± 0.24a	0.95 ± 0.01a	1.84 ± 0.06a	4.06 ± 0.70a	11.31 ± 1.6ef	10.86 ± 1.4 abc	10.51 ± 1.6 abc	
80%IR	4.53 ± 0.35cd	3.28 ± 0.04fg	4.47 ± 0.1cd	29.37 ± 0.80a	16.47 ± 0.20de	19.03 ± 1.40c	4.72 ± 1.02c	3.24 ± 0.40de	5.30 ± 0.04b	0.92 ± 0.00ef	1.72 ± 0.06b	2.96 ± 0.02b	6.58 ± 0.2d	7.91 ± 0.8de	7.90 ± 0.5d	
70%IR	3.47 ± 0.34f	5.19 ± 0.3b	7.11 ± 1.2a	16.61 ± 0.20ef	28.16 ± 0.30a	25.72 ± 0.90a	3.45 ± 0.40d	4.18 ± 1.60cd	4.70 ± 1.06bc	0.88 ± 0.02abc	1.64 ± 0.00bc	2.83 ± 0.04bc	5.55 ± 0.2ef	6.15 ± 0.2fg	6.59 ± 0.2e	
60%IR	2.97 ± 0.3gh	3.63 ± 0.42ef	5.01 ± 0.1b	14.75 ± 0.60fg	13.98 ± 0.90fg	15.07 ± 1.10de	2.43 ± 0.09ef	1.93 ± 0.21fg	3.40 ± 0.30de	0.85 ± 0.02bc	1.63 ± 0.00bc	2.73 ± 0.04cd	5.09 ± 0.08fgh	5.38 ± 0.1gh	6.09 ± 0.3ef	
50% IR	2.05 ± 0.29ij	4.49 ± 0.38de	3.87 ± 0.05e	12.24 ± 0.40h	14.13 ± 0.40fg	14.86 ± 0.90de	0.92 ± 0.11h	1.80 ± 0.40fg	3.00 ± 0.04e	0.83 ± 0.02cd	1.56 ± 0.00cd	2.66 ± 0.02d	3.11 ± 0.9j	4.66 ± 0.3hi	5.19 ± 0.1gh	
Akbar-2019	100%IR	5.15 ± 1.6c	4.07 ± 0.7de	4.04 ± 1.1de	22.67 ± 0.80c	15.03 ± 0.08ef	16.39 ± 0.50d	7.32 ± 2.00a	5.23 ± 0.40b	5.30 ± 0.40b	0.41 ± 0.04ef	1.23 ± 0.00e	2.10 ± 0.03f	15.08 ± 4.1a	14.98 ± 3.6a	10.84 ± 6.3abc	
80%IR	2.51 ± 0.1hi	3.98 ± 0.01ef	3.84 ± 1.14efg	9.48 ± 0.4i	12.95 ± 3.25gh	14.35 ± 0.90e	5.73 ± 0.15b	3.46 ± 0.34de	4.57 ± 0.48bc	0.27 ± 0.04gh	1.21 ± 0.02e	1.99 ± 0.02g	8.74 ± 0.3c	10.66 ± 0.9bc	11.10 ± 0.8ef	
70%IR	2.16 ± 0.14hi	2.91 ± 0.08h	3.43 ± 0.4f	9.11 ± 0.60i	10.90 ± 1.30hi	11.77 ± 0.30fg	1.64 ± 0.34fg	2.65 ± 0.20ef	3.58 ± 0.44de	0.23 ± 0.01hi	1.08 ± 0.07f	1.96 ± 0.00gh	7.73 ± 0.1cd	8.72 ± 0.5cd	9.32 ± 0.8cd	
60%IR	1.72 ± 0.2jk	1.84 ± 0.17j	3.12 ± 0.8g	12.74 ± 0.90h	8.29 ± 0.99j	11.87 ± 0.90fg	1.52 ± 0.47fg	3.27 ± 1.08de	3.52 ± 0.30de	0.22 ± 0.01i	1.02 ± 0.00fg	1.94 ± 0.00gh	7.13 ± 0.1d	7.68 ± 0.1de	8.66 ± 0.2d	
50% IR	1.44 ± 0.1k	1.34 ± 0.08jk	4.48 ± 0.3cd	8.40 ± 0.90j	10.51 ± 0.05hi	13.66 ± 2.75ef	0.55 ± 0.08h	2.54 ± 0.12ef	3.48 ± 0.50de	0.13 ± 0.01j	0.97 ± 0.00g	1.86 ± 0.01h	5.97 ± 1.2def	5.81 ± 1.4fg	6.25 ± 0.5ef	
FSD-08	100%IR	6.13 ± 0.84ef	3.49 ± 1.1f	4.39 ± 0.3cd	12.08 ± 0.50h	17.99 ± 0.50cd	18.99 ± 0.01c	3.50 ± 0.24d	4.69 ± 0.28bc	3.99 ± 0.90cd	0.73 ± 0.01de	1.34 ± 0.32d	2.58 ± 0.02e	11.34 ± 2.3ef	8.77 ± 1.5cd	9.15 ± 1.1cd	
80%IR	3.65 ± 0.88cd	2.17 ± 0.10i	3.28 ± 0.01fg	12.65 ± 0.60h	12.93 ± 0.30gh	15.43 ± 0.07de	2.41 ± 0.32ef	4.01 ± 0.44cd	3.69 ± 0.49de	0.60 ± 0.01fg	1.48 ± 0.00c	2.49 ± 0.03e	6.44 ± 0.2de	6.65 ± 0.8ef	7.28 ± 0.6de	
70%IR	2.05 ± 0.24i	1.74 ± 0.38j	3.17 ± 0.2g	15.58 ± 0.70f	12.42 ± 0.20gh	13.71 ± 0.20ef	2.01 ± 0.37f	4.03 ± 0.71cd	4.09 ± 0.60cd	0.65 ± 0.00ef	1.43 ± 0.00c	2.38 ± 0.04ef	5.79 ± 0.2ef	5.17 ± 0.4hi	6.11 ± 0.2ef	
60%IR	1.52 ± 0.2jk	1.51 ± 0.2jk	2.70 ± 0.2h	9.56 ± 0.70i	10.77 ± 0.05hi	11.87 ± 3.01fg	1.52 ± 0.47fg	3.27 ± 1.08de	4.49 ± 0.60bc	0.54 ± 0.00f	1.40 ± 0.02d	2.21 ± 0.05f	4.28 ± 0.1hi	4.59 ± 0.1hi	5.57 ± 0.2fg	
50% IR	1.26 ± 0.2k	1.89 ± 0.4j	2.68 ± 0.5hi	7.53 ± 0.20j	10.51 ± 0.60hi	12.55 ± 0.05fg	0.92 ± 0.11h	2.53 ± 0.56ef	2.81 ± 0.05e	0.48 ± 0.04fg	1.36 ± 0.02d	2.14 ± 0.00fg	3.36 ± 0.5ij	4.07 ± 0.3i	4.73 ± 0.5h	
Notes.

Data presenting the mean (n = 3) ± Standard deviation. Various lowercase letter superscripts indicate statistical significance at 95% confidence interval. IR indicates irrigation regimes, culti. represents cultivars, and AAB indicates activated acacia biochar.

Table 3 Means comparison for the effect of varying irrigation regimes, AAB and cultivars on root dry weight (g), plant height (cm), root length (cm), number of leaves, and number of tillers of wheat.

		Root dry weight (g)	Plant height (cm)	Root length (cm)	Number of leaves	Number of tillers	
Culti.	IR	0T-AAB	5T-AAB	10T-AAB	0T-AAB	5T-AAB	10T-AAB	0T-AAB	0T-AAB	0T-AAB	0T-AAB	5T-AAB	10T-AAB	0T-AAB	5T-AAB	10T-AAB	
Dilkash-2020	100%IR	1.75 ± 0.12cd	2.42 ± 0.04bc	2.44 ± 0.1bc	114.13 ± 0.49e	118.13 ± 1.33b	121.36 ± 2.23a	15.80 ± 0.80b	14.30 ± 0.20bc	15.30 ± 0.20b	27.66 ± 1.5c	29.33 ± 2.51b	32.66 ± 3.51a	2.66 ± 1.15d	3.66 ± 0.35c	6.00 ± 1.00a	
80%IR	1.24 ± 0.12fg	2.23 ± 0.03bc	2.30 ± 0.2bc	112.80 ± 0.40ef	115.13 ± 1.59d	118.26 ± 0.50b	14.60 ± 0.10bc	13.53 ± 0.40c	14.53 ± 0.40bc	23.33 ± 0.5cd	25.00 ± 0.00bc	28.66 ± 1.5b	2.66 ± 0.57d	3.66 ± 0.5c	5.66 ± 0.57ef	
70%IR	1.04 ± 0.04fgh	1.90 ± 0.10cd	1.94 ± 0.04cd	110.03 ± 1.65fg	111.56 ± 1.51ef	117.30 ± 0.45c	12.93 ± 0.23bc	11.63 ± 0.86bcd	12.63 ± 0.86bcd	20.33 ± 1.5de	23.66 ± 1.5cd	25.00 ± 0.0bc	4.00 ± 1.00bc	3.00 ± 0.00c	4.33 ± 0.57bc	
60%IR	0.70 ± 0.16gh	1.49 ± 0.20ef	1.62 ± 0.12de	104.40 ± 3.61fgh	108.46 ± 1.24efg	114.50 ± 0.79 e	10.26 ± 1.06d	10.50 ± 0.10d	11.50 ± 1.00cd	17.33 ± 1.5efg	19.00 ± 1.00def	22.00 ± 2.6cd	4.33 ± 0.57bc	2.66 ± 0.57d	4.66 ± 0.57bc	
50% IR	0.46 ± 0.05ghi	1.24 ± 0.02fg	1.41 ± 0.02ef	95.60 ± 2.25ghi	103.96 ± 2.30gh	111.33 ± 3.52ef	6.50 ± 1.60gh	9.00 ± 0.78de	10.00 ± 0.78d	13.00 ± 0.00ghi	15.33 ± 1.5gh	16.66 ± 1.5fg	3.66 ± 0.57c	3.00 ± 0.00c	4.33 ± 0.57bc	
Akbar-2019	100%IR	2.71 ± 0.25ef	2.63 ± 0.20ef	3.10 ± 0.12a	108.76 ± 3.51efg	110.10 ± 3.55fg	112.50 ± 4.1ef	19.40 ± 2.02a	14.16 ± 0.98b	15.70 ± 1.03b	22.66 ± 1.15cd	25.33 ± 1.15bc	28.66 ± 2.1b	5.33 ± 0.57ef	3.33 ± 1.15c	3.00 ± 0.00c	
80%IR	2.08 ± 0.02cd	2.16 ± 0.25c	2.76 ± 0.17ef	102.80 ± 0.70fgh	103.60 ± 0.96gh	105.73 ± 0.41gh	14.06 ± 0.72bc	11.73 ± 0.47bcd	13.50 ± 0.87bc	18.66 ± 1.15ef	20.66 ± 1.15de	21.66 ± 1.2cde	3.00 ± 1.00bc	4.33 ± 1.52bc	5.00 ± 0.00ef	
70%IR	1.73 ± 0.22cd	1.49 ± 0.06ef	2.25 ± 0.34bc	100.86 ± 0.75gh	101.50 ± 0.45gh	104.86 ± 0.15gh	10.93 ± 0.72d	10.36 ± 0.92d	12.00 ± 0.20bcd	17.00 ± 1.00efg	18.33 ± 1.15ef	20.66 ± 0.6de	2.66 ± 0.56cd	3.00 ± 0.00c	4.00 ± 0.00bc	
60%IR	1.22 ± 0.04fg	1.08 ± 0.07fgh	1.70 ± 0.18cd	98.23 ± 0.86ghi	100.30 ± 0.45gh	103.96 ± 0.25gh	9.03 ± 1.41de	8.70 ± 0.10ef	9.70 ± 0.10de	13.00 ± 1.7ghi	16.66 ± 0.00fg	18.33 ± 0.56ef	3.33 ± 0.57c	3.00 ± 1.00c	3.33 ± 0.5c	
50% IR	0.65 ± 0.04gh	0.61 ± 0.07gh	1.50 ± 0.02 cdf	92.20 ± 4.2ghi	94.76 ± 4.47ghi	100.43 ± 3.42gh	6.30 ± 0.66gh	7.16 ± 0.94fg	7.56 ± 0.80 egf	12.00 ± 0.00 i	14.33 ± 2.08gh	14.00 ± 2.0gh	3.33 ± 0.57c	4.00 ± 1.00bc	3.66 ± 1.15c	
FSD-08	100%IR	3.66 ± 1.00a	2.62 ± 0.24ef	2.65 ± 0.22a	109.30 ± 2.21fg	111.00 ± 2.45ef	112.93 ± 1.04ef	17.93 ± 0.40ef	13.96 ± 0.45c	15.03 ± 0.35ef	25.66 ± 3.7bc	27.66 ± 4.7c	30.00 ± 4.5ef	3.33 ± 0.5c	3.33 ± 1.15c	4.33 ± 1.15bc	
80%IR	2.87 ± 0.07ef	1.73 ± 0.17cd	2.30 ± 0.22ef	102.53 ± 1.76fgh	104.10 ± 2.13gh	108.20 ± 3.21efg	13.26 ± 1.58c	12.43 ± 0.41bcd	13.83 ± 0.70c	22.33 ± 0.5cd	24.00 ± 0.00bc	22.66 ± 2.5cd	3.33 ± 1.5c	3.66 ± 0.5c	4.33 ± 0.5bc	
70%IR	0.91 ± 0.07fgh	1.33 ± 0.13efg	1.96 ± 0.05cd	95.33 ± 2.30ghi	101.46 ± 0.94fgh	102.30 ± 0.79gh	10.93 ± 0.70d	11.10 ± 0.20cd	12.10 ± 0.20bcd	19.33 ± 1.5 e	20.88 ± 1.5de	20.33 ± 0.6de	4.66 ± 0.5bc	4.33 ± 0.5bc	4.66 ± 0.5bc	
60%IR	0.58 ± 0.06ghi	0.99 ± 0.06gh	1.77 ± 0.13d	92.13 ± 0.86ghi	96.50 ± 0.43ghi	98.67 ± 0.70ghi	8.60 ± 1.50ef	8.90 ± 0.70ef	9.80 ± 0.56de	16.33 ± 1.5fg	17.33 ± 2.1efg	18.33 ± 1.5ef	4.00 ± 0.00b	3.33 ± 0.5c	3.66 ± 0.5c	
50% IR	0.28 ± 0.04 i	0.73 ± 0.17gh	1.57 ± 0.03ef	89.46 ± 0.90hi	94.70 ± 0.43ghi	95.26 ± 1.00ghi	6.30 ± 0.50gh	6.76 ± 0.89 gf	8.30 ± 0.95ef	12.00 ± 2.6 i	13.00 ± 2.0ghi	14.66 ± 1.5gh	3.66 ± 1.15c	4.00 ± 1.00bc	3.66 ± 0.5c	
Notes.

Data presenting the mean (n = 3) ± Standard deviation. Various lowercase letter superscripts indicate statistical significance at 95% confidence interval. IR indicates irrigation regimes, culti. represents cultivars, and AAB indicates activated acacia biochar.

Table 4 Means comparison for the effect of varying irrigation regimes, AAB and cultivars on spike length (cm), spike weight (g), number of spikes/plant, number of grains/spike, number of spikelet/spikes, 1000-grain weight (g) of wheat.

		Spike length (cm)	Spike weight (g)	Number of spikes/plants	Number of grains/spikes	Number of spikelet/spikes	1000-Grain weight (g)	
Cult.	IR	0T-AAB	5T-AAB	10T-AAB	0T-AAB	5T-AAB	10T-AAB	0T-AAB	5T-AAB	10T-AAB	0T-AAB	5T-AAB	10T-AAB	0T-AAB	5T-AAB	10T-AAB	0T-AAB	5T-AAB	10T-AAB	
Dilkash-2020	100%IR	13.5 ± 1.3cd	14.5 ± 1.32bc	19.2 ± 1.36a	12.6 ± 2b	14.8 ± 2.9ef	16.8 ± 2.2a	5.0 ± 0.0b	6.0 ± 0.00ef	6.3 ± 0.5a	68.7 ± 1.5b	59.3 ± 1.84d	70.3 ± 1.5ef	21.6 ± 0.5bc	21.6 ± 0.5bc	23.0 ± 0.0ef	54.4 ± 0.9bc	56.4 ± 0.8b	61.5 ± 1.2a	
80%IR	12.1 ± 0.8cd	13.1 ± 0.8cd	17.7 ± 0.30b	8.7 ± 0.3ef	10.7 ± 0.9cd	11.1 ± 0.82bc	3.3 ± 0.5c	5.0 ± 0.00b	3.7 ± 0.5bc	64.0 ± 1.0cd	56.3 ± 1.9d	63.7 ± 1.5cd	20.0 ± 0.0bcd	19.3 ± 1.7d	19.0 ± 0.0d	49.7 ± 1.4de	50.6 ± 0.4d	55.9 ± 1.9bc	
70%IR	15.1 ± 1.8 abc	16.2 ± 1.8ef	17.4 ± 1.5b	7.7 ± 0.1fg	8.7 ± 0.5ef	9.3 ± 0.2de	2.0 ± 0.0d	4.3 ± 0.5bc	5.0 ± 0.00b	60.3 ± 0.5cde	72.6 ± 1.1a	68.3 ± 0.5b	19.0 ± 1.0d	21.0 ± 0.0bc	24.3 ± 0.5a	45.4 ± 0.9fg	51.9 ± 0.8cd	56.3 ± 0.1b	
60%IR	11.2 ± 1.1de	12.2 ± 1.1cd	14.0 ± 0.5bc	7.1 ± 0.1fg	7.7 ± 0.18fg	8.6 ± 0.5ef	2.0 ± 0.0d	3.3 ± 0.5c	3.3 ± 0.5c	58.3 ± 0.5d	67.0 ± 3.0c	62.7 ± 0.5cd	16.6 ± 1.1efg	18.0 ± 0.0de	17.0 ± 0.0ef	42.5 ± 0.4gh	49.6 ± 0.2de	50.3 ± 0.9d	
50% IR	11.7 ± 1.1de	12.8 ± 1.1cd	13.6 ± 0.3cd	5.6 ± 1.2hi	5.9 ± 0.1hi	6.4 ± 1.2gh	2.0 ± 0.0d	2.3 ± 0.5cd	3.0 ± 0.5c	56.0 ± 1.0de	59.7 ± 0.5d	60.3 ± 0.5cde	16.3 ± 0.5efg	16.6 ± 1.1efg	19.6 ± 1.5d	39.9 ± 0.5ghi	47.3 ± 0.4def	47.9 ± 0.6def	
Akbar-2019	100%IR	11.5 ± 1.4de	12.5 ± 1.4cd	14.2 ± 2.2bc	10.3 ± 1.4cd	11.1 ± 1.5bc	11.3 ± 2.3bc	3.7 ± 1.5bc	5.0 ± 0.00b	4.3 ± 0.5bc	65.0 ± 1.0bc	57.0 ± 1.0d	65.7 ± 0.5bc	19.0 ± 0.0d	20.3 ± 1.1bcd	19.7 ± 3.2d	52.7 ± 0.5bcd	51.1 ± 1.3cd	53.2 ± 1.0bc	
80%IR	11.7 ± 0.7de	12.7 ± 0.7cd	10.9 ± 0.5ef	6.6 ± 0.2gh	7.8 ± 0.6fg	8.0 ± 0.6ef	3.3 ± 0.5c	3.7 ± 0.5bc	3.0 ± 0.00c	62.8 ± 0.5c	66.7 ± 1.5bc	60.7 ± 0.5cde	16.7 ± 0.5efg	18.0 ± 0.0de	18.3 ± 3.2d e	48.7 ± 1.3de	48.7 ± 1.3de	48.6 ± 0.6def	
70%IR	10.5 ± 0.1ef	11.5 ± 0.8de	11.4 ± 0.2de	5.5 ± 0.2hi	6.2 ± 0.2gh	6.6 ± 0.2gh	2.3 ± 0.5cd	4.0 ± 0.0bc	4.0 ± 0.00bc	59.7 ± 1.5de	62.0 ± 1.0cd	61.7 ± 0.5cde	15.7 ± 0.5fg	21.0 ± 0.0bc	19.6 ± 4.0d	41.4 ± 1.0gh	50.4 ± 0.2d	45.7 ± 1.3fg	
60%IR	10.3 ± 0.1ef	11.3 ± 0.1de	11.7 ± 0.6de	5.1 ± 0.1hi	5.3 ± 0.1hi	6.1 ± 0.2gh	2.0 ± 0.0d	3.0 ± 0.0c	3.0 ± 0.00c	55.7 ± 1.5de	58.0 ± 1.7de	57.7 ± 1.5de	14.3 ± 0.5gh	17.0 ± 0.0ef	15.3 ± 0.5fg	39.3 ± 0.5ghi	47.0 ± 1.7def	44.4 ± 0.8fg	
50% IR	11.2 ± 1.4de	10.1 ± 0.7ef	10.8 ± 0.8ef	3.1 ± 0.9j	4.3 ± 0.1ij	4.5 ± 0.85ij	1.3 ± 0.5de	2.3 ± 0.5cd	3.0 ± 1.00c	52.7 ± 0.5efg	53.7 ± 0.5ef	53.3 ± 1.5ef	13.0 ± 0.0gh	16.0 ± 0.0efg	15.6 ± 1.1fg	37.1 ± 0.9hi	44.1 ± 0.6fg	43.5 ± 0.5fgh	
FSD-08	100%IR	11.6 ± 1.3de	12.6 ± 1.3cd	10.2 ± 0.8ef	8.7 ± 1.5ef	9.1 ± 1.1de	11.3 ± 2.3bc	4.3 ± 0.5bc	4.0 ± 0.0bc	4.0 ± 1.00bc	64.0 ± 2.0c	63.7 ± 2.0cd	63.0 ± 2.0cd	20.0 ± 0.0bcd	20.3 ± 0.5bc	18.0 ± 0.0de	52.0 ± 1.6bcd	52.9 ± 0.6bcd	50.7 ± 0.5d	
80%IR	9.9 ± 0.7fg	10.9 ± 0.7ef	13.4 ± 3.1bc	6.3 ± 0.3gh	6.8 ± 0.7gh	7.3 ± 0.6fg	4.0 ± 0.0bc	3.0 ± 0.0cd	3.3 ± 0.5c	61.3 ± 1.5cd	53.0 ± 1.0ef	55.7 ± 2.0de	18.0 ± 0.0de	18.6 ± 0.5de	15.6 ± 1.5fg	46.7 ± 3.6efg	49.2 ± 1.1de	47.3 ± 0.2ef	
70%IR	9.6 ± 0.6fg	10.6 ± 0.6ef	11.4 ± 0.2de	5.2 ± 0.4hi	5.7 ± 0.2hi	6.11 ± 0.2gh	3.0 ± 0.0c	4.0 ± 0.0bc	4.3 ± 0.5bc	57.0 ± 1.0de	57.7 ± 4.0de	61.0 ± 1.0cde	16.0 ± 0.0efg	22.0 ± 0.0ef	20.0 ± 0.0bc	42.5 ± 1.1gh	50.4 ± 0.8d	44.9 ± 0.5fgh	
60%IR	9.8 ± 0.6fg	10.9 ± 0.6ef	10.8 ± 0.8ef	4.3 ± 0.1ij	4.5 ± 0.1ij	5.6 ± 0.23hi	3.0 ± 0.0c	2.3 ± 0.5cd	2.7 ± 0.5cd	52.3 ± 0.5efg	55.6 ± 2.8de	53.0 ± 1.0ef	14.0 ± 0.0gh	17.3 ± 0.5ef	16.0 ± 0.0efg	39.3 ± 0.8ghi	45.6 ± 0.9fg	43.4 ± 0.5fgh	
50% IR	9.1 ± 1.3g	10.2 ± 0.7ef	9.6 ± 0.7efg	3.4 ± 0.5j	4.2 ± 0.1ij	4.5 ± 0.8ij	2.0 ± 0.0d	2.7 ± 0.5cd	2.7 ± 0.5cd	51.0 ± 1.0fg	52.0 ± 1.0rfg	52.0 ± 1.0rfg	12.7 ± 0.5i	16.7 ± 0.5efg	15.0 ± 0.0fg	35.0 ± 3.1i	41.1 ± 0.6ghi	42.1 ± 0.6gh	
Notes.

Data presenting the mean (n = 3) ± standard deviation. Various lowercase letter superscripts indicate statistical significance at 95% confidence interval. IR indicates irrigation regimes, culti. represents cultivars, and AAB indicates activated acacia biochar.

Figure 5 Mean comparison for the effect of varying irrigation regimes, AAB and cultivars on (A) seed yield per hectare (kg ha−1) and (B) apparent water productivity (kg m−3) of wheat.

Where alphabets indicate statistical significance at 95% confidence interval, 0T, 0 tons per hectare; 5T, 5 tons per hectare; 10T, 10 tons per hectare; AAB, activated Acacia biochar; and IR, Irrigation regime; Dilkash, Dilkash-2020 cultivar; Akbar, Akbar-2019 cultivar; and FSD-08, FSD-08 cultivar.

Pearson correlation

Pearson correlation among all the observed traits was performed to understand their correlation with the most relevant traits (Fig. 6). Wheat morphological attributes as well as yield attributes were positively correlated. As the red color in the plot presents a positive association between two traits and blue shows a negative correlation, whereas the white color shows no correlation among traits. It was observed that carotenoids, MDA, proline, POD, and SOD significantly had a negative correlation with all other traits.

Figure 6 Pearson correlation for interaction among observed traits for all cultivars of wheat under deficit irrigation with AB amendments under varying IR levels (* significance at p ≤0.05).

PH, plant height; SL, spike length; RL, root length; NOL, number of leaves; NOT, number of tillers; RWC, relative water content; MSI, membrane stability index; ChlA, chlorophyll a; ChlB, chlorophyll b; CAT, catalase; POD, peroxidase; SOD, superoxide dismutase; MDA, malondialdehyde; SPDW, spike dry weight; NGPS, number of grain per spike; NGPP, number of grain per plant; NSPP, number of spikes per plant; TGW, thousand grain weight, number of spikelet per spike, LDW, leaf dry weight; SDW, stem dry weight; RDW, root dry weight; YP, yield per hectare; AWP, apparent water productivity; LFW, leaf fresh weight; SFW, stem fresh weight; RFW, root fresh weight.

Multivariate analysis

This study utilizes principal component analysis (PCA) heatmap and biplot to explore the relationship between activated acacia biochar amended soil under varying irrigation regimes cultivated with different wheat cultivars and physiological, biochemical, and yield variables (Fig. 7). The analysis effectively distinguished plants exposed to deficit irrigation with activated biochar amended and non-amended soil. The heatmap clearly showed that the changes made by 10T-AAB had the most significant impacts on morphological, physiological, and yield attributes (Fig. 8A). The most prominent effect among morphological traits with 10T-AAB level of soil amendment were observed in number of leaves, plant height, and plant fresh and dry weights. Whereas among physiological traits, a significant effect of activated biochar amendment was observed for plant relative water content, membrane stability index, chlorophyll contents, and antioxidants under deficit irrigations. However, 0T-AAB had the highest levels of proline content, catalase, peroxidase, and superoxide dismutase, indicating the severity of stress. The biplot presented two main clusters each representing a group of applied treatments (Fig. 8B).

Figure 7 Principal component analysis (PCA) Biplot showing the effect of varying irrigation regimes, AB and cultivars on plant responses.

PH, plant height; SL, spike length; RL, root length; NOL, number of leaves; NOT, number of tillers; RWC, relative water content; MSI, membrane stability index; ChlA, chlorophyll a; ChlB, chlorophyll b; CAT, catalase; POD, peroxidase; SOD, superoxide dismutase; MDA, malondialdehyde; PRO, protein; Pro, proline; SPDW, spike dry weight; NGPS, number of grain per spike; NGPP, number of grain per plant; NSPP, number of spikes per plant; TGW, thousand grain weight, number of spikelet per spike; LDW, leaf dry weight; SDW, stem dry weight; RDW, root dry weight; YP, yield per hectare; AWP, apparent water productivity; LFW, leaf fresh weight; SFW, stem fresh weight; RFW, root fresh weight. 1 = 0T AAB+ 100% IR+ Dilkash, 2 = 0T AAB+ 80% IR+ Dilkash, 3 = 0T AAB+ 70% IR+ Dilkash, 4 = 0T AAB+ 60% IR+ Dilkash, 5 = 0T AAB+ 50% IR+ Dilkash, 6 = 0T AAB+ 100% IR+ Akbar, 7 = 0T AAB+ 800% IR+ Akbar, 8 = 0T AAB+ 70% IR+ Akbar, 9 = 0T AAB+ 600% IR+ Akbar, 10 = 0T AAB+ 500% IR+ Akbar, 11 = 0T AAB+ 100% IR+ FSD-08, 12 = 0T AAB+ 80% IR+ FSD-08, 13 = 0T AAB+ 70% IR+ FSD-08, 14 = 0T AAB+ 60% IR+ FSD-08, 15 = 0T AAB+ 50% IR+ FSD-08, 16 = 5T AAB+ 100% IR+ Dilkash, 17 = 5T AAB+ 80% IR+ Dilkash, 18 = 5T AAB+ 70% IR+ Dilkash, 19 = 5T AAB+ 60% IR+ Dilkash, 20 = 5T AAB+ 50% IR+ Dilkash, 21 = 5T AAB+ 100% IR+ Akbar, 22 = 5T AAB+ 800% IR+ Akbar, 23 = 5T AAB+ 70% IR+ Akbar, 24 = 5T AAB+ 600% IR+ Akbar, 25 = 5T AAB+ 500% IR+ Akbar, 26 = 5T AAB+ 100% IR+ FSD-08, 27 = 5T AAB+ 80% IR+ FSD-08, 28 = 5T AAB+ 70% IR+ FSD-08, 29 = 5T AAB+ 60% IR+ FSD-08, 30 = 5T AAB+ 50% IR+ FSD-08, 31 = 10T AAB+ 100% IR+ Dilkash, 32 = 10T AAB+ 80% IR+ Dilkash, 33 = 10T AAB+ 70% IR+ Dilkash, 34 = 10T AAB+ 60% IR+ Dilkash, 35 = 10T AAB+ 50% IR+ Dilkash, 36 = 10T AAB+ 100% IR+ Akbar, 37 = 10T AAB+ 800% IR+ Akbar, 38 = 10T AAB+ 70% IR+ Akbar, 39 = 10T AAB+ 600% IR+ Akbar, 40 = 10T AAB+ 500% IR+ Akbar, 41 = 10T AAB+ 100% IR+ FSD-08, 42 = 10T AAB+ 80% IR+ FSD-08, 43 = 10T AAB+ 70% IR+ FSD-08, 44 = 10T AAB+ 60% IR+ FSD-08, 45 = 10T AAB+ 50% IR+ FSD-08, where Dilkash = Dilkash-2020 cultivar, Akbar = Akbar-2019 cultivar, and FSD-08 = FSD-O8 cultivar.

Figure 8 Heatmap showing effective significance of AAB under varying irrigation regimes on wheat cultivars

B0 = 0T-AAB, B1 = 5T-AAB, B2 = 10T-AAB, D0 = 100% IR, D1 = 80% IR, D2 = 70% IR, D3 = 60% IR, D4 = 50% IR, V1 = Dilkash-2020, V2 = Akbar-2019, and V3 = FSD-08.

Discussion

Among several types of biochar, the one produced using wood biomass exhibits a large surface area due to the higher lignin content of the wood. Further, its activation with organic wastes adds valuable and promising properties (Jahan et al., 2023). The current study showed that increased water holding capacity with activated biochar application is consistent with ameliorating soil health and fertility. Biochar improved soil water retention by improving the soil’s micropore structure and reducing the macropore surface (Abel et al., 2013). This reduction was attributed to biochar’s ability to fill macropore surfaces in soil, influencing pore size distribution and leading to higher soil density with increased micropore proportion. Moreover, biochar itself has a porous nature, which adds to the total soil pore volume, improving aerations and water infiltration (Karhu et al., 2011). Under biochar amendment, soil pH was increased from acidic to neutral, which can be a consequence of the liming effect of biochar due to the presence of basic cations such as magnesium, calcium, and potassium (Yuan & Xu, 2011). Variation in electrical conduction of soil with AAB amendment showed that biochar initially releases soluble salts, thereby increasing the electrical conductivity of soil, but over time, these salts are utilized by plants or leached away (Major et al., 2010). Moreover, carbon recovery, mean residence time, and carbon sequestration are a consequence of recalcitrant carbon components in acacia activated biochar that remain in soil over a longer period.

The levels of proline and lipid peroxidation increased in water deficit conditions. But biochar reduced these stress markers by improving soil water retention in current study which is in agreement with the findings of Gharred et al. (2022). Plants usually accelerate their antioxidant activity to cope with reactive oxygen species (ROS) produced under abiotic stress (Mu et al., 2021). The current study observed an increased level of protein content but reduced antioxidants (CAT, POD, and SOD) with biochar application under low irrigation regimes, especially 60% and 50%. This effect was attributed to biochar’s ability to enhance ROS scavenging mechanisms, reducing oxidative stress (Nawaz et al., 2023), which in turn provides protection against lipid damage (Farhangi-Abriz & Torabian, 2017), as can be evidenced by reduced proline, MDA contents, and antioxidant levels.

Sugar content, RWC, and MSI were enhanced with 10T-AAB even under low irrigation levels. Improved sugar contents were associated with reduced osmotic stress and enhanced soil fertility due to increased soil organic carbon (SOC). Higher SOC levels with biochar amendment lead to improved nutrient availability and metabolic activities (Jahan et al., 2023). Tanure et al. (2019) stated that biochar’s porous structure enhances water retention that helps maintain higher RWC in plant tissues, thereby improving cellular metabolism. Drought stress directly affects the photosynthetic ability of plants, which can be ameliorated using biochar (Sattar et al., 2019). Photosynthetic pigments were observed to be increased with increasing levels of AAB and played a role in mitigating the negative effects of a low irrigation regime. These increased levels with biochar were attributed to enhanced nutrients particularly nitrogen, which is crucial component of porphyrin ring of chlorophyll molecule, which in turn promoted chlorophyll biosynthesis (Abideen et al., 2020). Additionally, the study observes that biochar alters soil pH, influencing nutrient absorption and availability in the rhizosphere (Ayaz et al., 2021). The observed improved soil structure contributes to better plant development (Manolikaki & Diamadopoulos, 2019).

Plants treated with activated biochar showed enhanced root growth, which can be attributed to improved soil structure and increased nutrient availability (Zhao et al., 2019). According to Jahan et al. (2023), biochar is characterized by high surface area enhancing soil water holding capacity, which improves soil aeration and reduces soil compaction, allowing extensive root growth. As previously indicated, biochar can improve root growth in plants, which is essential for increasing water use efficiency and plant survival during dry spells (Zulfiqar et al., 2022; Tanure et al., 2019).

The present results demonstrated that biochar amendments contribute to reducing losses in wheat growth and yield by retaining water in soil pores and gradually releasing it under moisture deficit, like the findings of Ali et al. (2017). Activated biochar positively influenced spike development, which can be attributed to enhanced chlorophyll contents, relative water contents, protein accumulation, and increased soil nutrients especially carbon and nitrogen, essential for plant reproductive growth.

The number of spikelets per spike along with grain filling was higher in 10T-AAB treated plants leading to heavier grains as compared to control, which is according to the findings of Haider et al. (2020), indicating better grain quality and yield with biochar application even under low irrigation regimes (Zulfiqar et al., 2022). Hence, slow crop growth rates, poor source–sink relationships, and malfunctioning metabolic systems contributing to low grain weight under drought stress can be controlled with activated biochar as a soil amendment. However, with activated biochar application, interactive effects of plant growth attributes with soil physicochemical properties and plant’s physiological sustainability against oxidative stress cause resilience to water deficit condition and lead to increased wheat production. Thus, AAB improves stress resilience benefiting wheat growth and yield under water deficit conditions assuring food security.

Conclusion

The current study illustrates the potential of activated acacia biochar (AB) to enhance wheat crop productivity under deficit water conditions, as can be evidenced by improved apparent water productivity and overall crop yield. The application of AAB resulted in significant improvements in soil physicochemical properties, enhancing soil organic carbon, water holding capacity, electrical conductivity, hydrogen, oxygen, carbon recovery, and carbon sequestration capacity. Furthermore, AAB enhanced drought tolerance in wheat plants by improving biochemical contents, promoting root growth, and enhancing photosynthetic efficiency, as can be witnessed by enhanced photosynthetic pigments. This soil amendment also enhanced water availability to plants by improving water retention in soil micropores, increasing crop resilience to deficit irrigations. These positive effects ultimately translated into improved yield attributes, such as increased grain yield. Overall, the findings highlight the promising role of AAB as a sustainable agricultural practice for mitigating the adverse impacts of water scarcity on wheat cultivation. Further investigations are necessary to optimize application rates and assess long-term effects on soil-plant-water interactions and crop productivity. As activated acacia biochar plays a role in remediation of degraded soil health, hence, soil microbiome under the effect of biochar can be studied further. Moreover, there is need to integrate activated biochar in agricultural soils to combat deficit water conditions because of climate change which needs technical training of people associated with agricultural sectors especially farmers about the impact of activated biochar.

Supplemental Information

Supplemental Information 1 R Codes for PCA analysis and Heatmap analysis

Supplemental Information 2 Physiological, biochemical and yield attributes of wheat under varying irrigation regimes

Additional Information and Declarations

Competing Interests

Author Contributions

Data Availability

The authors declare there are no competing interests.

Lubaba Komal performed the experiments, analyzed the data, prepared figures and/or tables, authored or reviewed drafts of the article, and approved the final draft.

Summera Jahan conceived and designed the experiments, analyzed the data, authored or reviewed drafts of the article, achieved funding, and approved the final draft.

Atif Kamran analyzed the data, authored or reviewed drafts of the article, and approved the final draft.

Abeer Hashem analyzed the data, authored or reviewed drafts of the article, achieved funding, and approved the final draft.

Graciela Dolores Avila-Quezada analyzed the data, authored or reviewed drafts of the article, and approved the final draft.

Elsayed Fathi Abd_Allah analyzed the data, authored or reviewed drafts of the article, achieved funding, and approved the final draft.

The following information was supplied regarding data availability:

The codes used for multivariate analysis and raw measurements are available in the Supplementary Files.

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
