# Peer review of "Optimizing soil health through activated acacia biochar under varying irrigation regimes and cultivars for sustainable wheat cultivation"

_PeerJ, doi:10.7717/peerj.18748_

## Round 0.1 · original submission · Major Revisions

Dear Authors

The reviewers have recommended revisions to your manuscript. Therefore, I invite you to respond to the reviewers' comments and revise your manuscript.

In addition, there are significant concerns about the manuscript's grammar, usage, and overall readability. We, therefore, request that you revise the text to fix the grammatical errors and improve the overall readability of the text.

With Thanks

·

Basic reporting

The research is original, novel and free of plagiarism. The research is clear and unambiguous, simple and professional language was used. The introduction and review is aligned with the study. Figures are relevant to the study and high-resolution pics were prepared. The labelling of figures is appropriate; however, figure labels shall be at the bottom of figures. Please clearly define the study rationale and research gap. Please discuss how different moisture ranges were maintained? Properly discuss the results of PCA analysis and heatmap and trace a relationship between different doses of biochar and various physiological and biochemical parameters. Also, discuss the future perspective and any limitations of the study in the conclusion

Experimental design

The experimental design is correctly used to meet the objectives. The field experiments were carefully performed to solve the real time problems

Validity of the findings

The experiments were carefully designed and all the required parameters were recorded. The data analyses is appropriate and represents the true picture of experiments. Large sample size was used along with the complete details of environmental conditions and doses of biochar.

Additional comments

The experiments were carefully performed and this research paper will be very helpful for the readers to enhance their knowledge and design future experiments. Few grammatical mistakes have been highlighted in the manuscript in the form of track changes eg. Spelling mistakes, few references are incomplete. Figure captions shall be below the figures. Future prospects shall be clearly mentioned.

Reviewer 2 ·

Basic reporting

The manuscript entitled "Optimizing soil health through activated acacia biochar under varying irrigation regimes and cultivars for sustainable wheat cultivation" is an interesting study showing the effects of using biochar as organic fertilizers, under various irrigation levels on different cultivars of wheat. It could be a significant addition to the journal. However, the authors need to address a few queries mentioned below, that could be helpful in improving the manuscript.

In the Introduction section, add more recent literature, particularly related to biochar's application in specific cropping systems or field studies, to show the novelty of this research.

The hypothesis and objectives must be stated clearly at the end of section. Also add a statement highlighting the novelty of the study.

Check the grammatical and formatting errors throughout the manuscript.

Experimental design

Specify the application methods for biochar and fertilizers, timing, dosage and any maintenance requirements.

The description of irrigation regimes lacks clarity, especially in terms of defining field capacity for each treatment level (100%, 80%, etc.). Additional detail could be added, on how these were maintained throughout the season.

At few places, there are typing mistakes and inconsistent formatting, such as "biochar" was misspelled as "biocahr" and "Lutron PMS-714" not being clearly introduced as a soil moisture device.

Validity of the findings

In Results section, phrases like "compared to 0T-AAB" and "under deficit IR conditions" are repeated excessively.

Add some details regarding interpretation of results obtained from PCA and heatmap analyses.

Integrate the results of physiological and molecular analyses and represent the combined effects in conclusion section.

Add significant findings in conclusion section to make it more clear.

Reviewer 3 ·

Basic reporting

In the present study entitled "Optimizing soil health through activated acacia biochar under varying irrigation regimes and cultivars for sustainable wheat cultivation", authors found that wheat productivity was enhanced by the activated acacia biochar under water deficit irrigation. I think that the work falls into the scope of the journal and the findings are interesting, however MS demands major revision.
Comments:
Abstract: The abstract can be more concise. I would suggest adding numerosity associated with each parameter for a better understanding of the main outcomes and context of the research. abstract. Add/change some keywords.
Introduction: Authors should add the novelty of the research and conceive a strong idea from the review literature about limitations and Challenges to coping with research gaps to conceive strong idea from literature review highlighting a) Standardization of biochar properties b) long-term impacts on soil properties need more investigation while more studies focus on short-term impact on soil water retention c) little focus on physiological studies etc.

Experimental design

Materials and methods: How many replications per treatment? How many plants per replication? Day/light hrs? Temperature? The drafting of many sentences needs to be revised. Please standardize hour for hour/hours. Materials and methods require more information with proper instrument name and model. No doubt, the author provided some details among various sections, but I think it is better to check all sections.
Figures. I noticed many meteorological parameter graphs; in my opinion, all parameters should be combined into a single plot and coloured by line or graph.

Validity of the findings

Results and Discussion: In results, there is a connection between sentences and paragraphs, but I suggest restructuring the text to avoid making contradictory statements between results and discussion. There are many values in results that increase the ubiquity of results. I would suggest presenting your results by increase/decrease %age. The percentage should be up to two digits e.g., 28% instead of 27.93%. One way of improving Discussion is to avoid repetition of results in this part. In discussion, there is a lack of a mechanistic approach.
The conclusions should be supported by data on how they are linked to goals and how this information contributes to knowing the gaps.
Tables. The homogenous group should be added as a superscript with the mean value and SE mean to know if there is a significant difference among the treatments or not.

Additional comments

Spellings and the English language need to be checked thoroughly. Overall, the drafting of many sentences needs to be improved. Tidying up the text is also suggested.

---

## Round 0.2 · accepted · Accept

Dear Authors,
I am pleased to inform you that the manuscript has improved after the last revision and can be accepted for publication.
Congratulations on accepting your manuscript, and thank you for your interest in submitting your work to PeerJ.
With Thanks

·

Basic reporting

The authors have effectively addressed all the suggested revisions in the manuscript. I am satisfied with the modifications made, and the article is now suitable for acceptance and publication.

Experimental design

The experimental design has been elaborated with adequate details, ensuring clarity and reproducibility of the study.

Validity of the findings

The results appear to be valid and are supported by appropriate data and analysis.

Reviewer 2 ·

Basic reporting

Authors have addressed all the concerns effectively. The revised version of the manuscript demonstrates significant improvements and could be a valuable contribution to sustainable agriculture research. Authors have provided sufficient background and context with relevant and recent literature supporting their study. The manuscript can now further be processed.

Experimental design

The experimental design is appropriate for testing the effects of AAB and irrigation regimes on wheat cultivars.

Validity of the findings

The findings are well supported by the relevant literature.

Additional comments

Line 86-90, seems to be unnecessary, as some of the text has already been mentioned above in the section.

Reviewer 3 ·

Basic reporting

Title
The title clearly describes the article.

Abstract
The abstract is well structured.

Introduction
The introduction is up to mark of scientific background and highlights the updated literature review thus representing aims and objectives.

Experimental design

Materials and Methods
The methodology is well written and all components are clearly described now.

Validity of the findings

The author significantly improved a large part of the language to ensure meaningful sentences.

Additional comments

The authors responded positively to most of the comments.